# JaxPlan and GurobiPlan: Optimization Baselines for Replanning in Discrete and Mixed Discrete and Continuous Probabilistic Domains

**Primary Keywords:** *None*

## Abstract

Replanning methods that determinize a stochastic planning problem and replan at each action step have long been known to provide strong baseline (and even competition winning) solutions to discrete probabilistic planning problems. Recent work has explored the extension of replanning methods to the case of mixed discrete and continuous probabilistic domains by leveraging MILP compilations of the RDDL specification language. Other recent advances in probabilistic planning have explored the compilation of structured mixed discrete and continuous RDDL domains into a determinized computation graph that also lends itself to replanning via so-called planning by backpropagation methods. However, to date, there has not been any comprehensive comparison of these recent optimization-based replanning methodologies to the state-of-the-art winner of the discrete probabilistic IPC 2011 and 2014 and runner-up in 2018 (PROST) and the winner of the mixed discrete-continuous probabilistic IPC 2023 (DiSProd). In this paper, we provide JaxPlan that has several extensive upgrades to both planning by backpropagation and its compact tensorized compilation from RDDL to a Jax computation graph with discrete relaxations and a sample average approximation. We also provide the first detailed overview of a compilation of the RDDL language specification to Gurobi's Mixed Integer Nonlinear Programming (MINLP) solver that we term GurobiPlan. We provide a comprehensive comparative analysis of JaxPlan and GurobiPlan with competition winning planners on 19 domains and a total of 155 instances to assess their performance across (a) different domains, (b) different instance sizes, and (c) different time budgets. We also release all code to reproduce the results along with the open-source planners we describe in this work.

## Introduction

Stochastic planning addresses decision-making under uncertainty subject to probabilistic state transitions, and plays a major role in diverse fields such as robotics, artificial intelligence, and operations research. While historical emphasis in stochastic planning centered on discrete problems, recent years have witnessed a growing interest in continuous and mixed discrete-continuous problems due to their ability to accurately model a wide range of real-world scenarios with continuous time, space, and resources (Li and Williams 2011; Fernandez-Gonzalez, Williams, and Karpas 2018).

The International Planning Competition (IPC) (McDermott 2000; Coles et al. 2012; Vallati et al. 2015), particularly its probabilistic track (IPPC), serves as a pivotal platform for evaluating stochastic planning systems. Replanning methods, historically prominent in control (Morari, Garcia, and Prett 1988) and planning (Hoffmann and Nebel 2001), have demonstrated surprising efficacy in discrete probabilistic planning competitions. Notably, FF-Replan (Yoon, Fern, and Givan 2007), the winner of the first IPPC in 2004, efficiently utilizes the deterministic planner FF (Hoffmann and Nebel 2001) to obtain plans based on determinized problem versions, effectively adapting to unexpected events during execution by replanning after each action is taken and the stochastically sampled next state is observed. In response to suggestions that all probabilistic planning might be effectively reduced to determinized replanning, Little and Thiebaux (2007) introduced a class of problems referred to as *Probabilistically Interesting* that could lead to suboptimal deterministic replanning behavior; this work influenced domain design in subsequent IPPCs as we highlight later.

Since 2011, RDDL (Sanner et al. 2010) has been the standard description language for IPPCs, replacing PPDDL (Younes et al. 2005). RDDL has the capacity to represent discrete and continuous components, with concurrency and exogenous and endogenous noise, while PPDDL is limited to only endogenous noise caused by stochastic action effects, and only a single action per time step. The IPPCs of 2011, 2014, and 2018 introduced only discrete problems and thus did not utilize the full expressive range of RDDL.

The winners of the IPPCs over the years used a variety of methods, mainly revolving around the idea of replanning after each action is taken. PROST (Keller and Eyerich 2012), the winner of IPPC 2011 and IPPC 2014, is a UCT based search method that leverages the structure of the problem, and knows to ignore actions if their effect does not have impact. SOGBOFA (Cui and Khardon 2016), the runner up of IPPC 2018, is a symbolic gradient-based optimization method, that symbolically represents an approximation of the Q value function as a function of the action variables in order to perform gradient-based search over actions. DiSProD (Chatterjee et al. 2023), the winner of IPPC 2023 mixed discrete-continuous competition, is a replanning method able to handle mixed problems by approximating the discrete components with a second order Taylor expansion. DiSProD is able to perform a gradient search forward and apply the first action in a rolling horizon scheme.

The challenge of mixed discrete-continuous domains has been persistent for probabilistic planners, prompting dedicated focus in the 2023 IPPC. While determinization and replanning worked well in the discrete competition, it was interesting to see if these schemes are also effective in the mixed discrete-continuous case. The 2023 IPPC also introduced JaxPlan as a baseline method, evaluated in both straight-line planning mode (without replanning) and deep reactive policy mode.

In this study, we evaluate PROST and DiSProD on the domains they previously won, alongside JaxPlan in a replanning mode with a comprehensive method description. Additionally, we introduce for the first time GurobiPlan, a novel replanning method based on compiling RDDL descriptions into mathematical optimization problems. Both GurobiPlan and JaxPlan undergo evaluation across all domains, providing insights into their capabilities in both purely discrete and mixed discrete-continuous domains.

One of our core objectives in this paper is to establish replanning baselines for mixed discrete-continuous problems and to further evaluate how well they work in purely discrete scenarios. It is noteworthy that FF-Replan is excluded from our evaluation here due to its specificity to PPDDL and challenges in adapting to RDDL domains featuring concurrency or continuous components; this is attributed to the unbounded number of potential outcomes in the lifted RDDL specification (cf. RDDL domains like Wildfire) that PPDDL cannot model. Additionally, while the 2018 IPPC (Geißer 2019) introduced other PROST variants and methods such as SOGBOFA (Cui and Khardon 2016), it introduced a variant of RDDL that has not been supported in the 2023 IPPC, which precludes comparative evaluation with JaxPlan or GurobiPlan that is the focus of this work.

## Background

**Markov Decision Process** A Markov decision process (MDP) is a tuple $(\mathcal{S}, \mathcal{A}, P, r, \gamma)$, where $\mathcal{S}$ is the state space, $\mathcal{A}$ is the action space, $P$ is the next-state distribution, $r$ is the single step reward function and $\gamma$ is the discount factor. A plan is a sequence of actions $a_1, \ldots a_T$, while a policy $\pi$ is a mapping from states to actions. The goal of probabilistic planning is to find a sequence of actions that maximizes the cumulative expected reward over a planning horizon $T$

$$\max_{a_1 \ldots a_T} \mathbb{E}_{s_t \sim P(s_{t-1}, a_{t-1}, \cdot)} \left[ \sum_{t=1}^{T} r(s_t, a_t) \,\Big|\, s_1 = s \right]. \quad (1)$$

Since the optimization problem above can be computationally intractable for long horizon, replanning approaches typically use a much shorter receding horizon $T' \ll T$ and solve (1) periodically starting from the current state $s_t$ at every decision epoch (or more generally, every $k$ epochs).

**Planning-By-Backpropagation** (PBBP) compiles a planning problem into a unrolled differentiable computational graph (Schulman et al. 2015), thus enabling the direct calculation of return gradients with respect to action-fluents or policy parameters (Wu, Say, and Sanner 2017; Bueno et al. 2019; Patton et al. 2022) as illustrated conceptually in Figure 1. Consider first a deterministic MDP

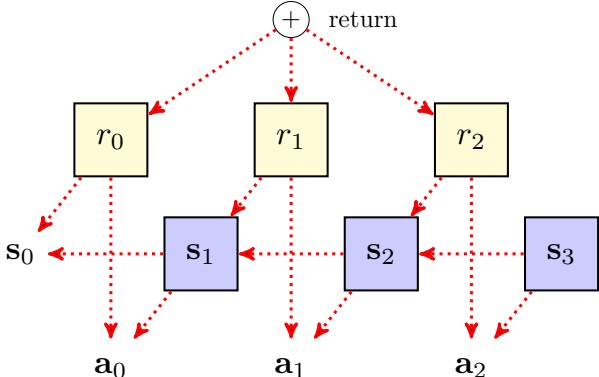

Figure 1: A differentiable computation graph for an abstract MDP with a horizon of 3 decision time steps. Here, $\mathbf{s}_t$ and $\mathbf{a}_t$ denote the state and action variables, respectively. Red dotted arrows indicate the flow of gradients from the return objective $\sum_t r_t$ with respect to actions in the computation graph.

with state transition model $s_{t+1} = f(s_t, a_t)$ and reward $r(s_t, a_t)$, where $f$ and $r$ are both differentiable functions. Trajectories $s_1, s_2 \ldots s_T$ can be forward sampled directly from the model, and the gradients of the return $\nabla_{a_1, \ldots a_T} V(a_1, \ldots a_T, s_1) = \nabla_{a_1, \ldots a_T} \sum_{t=1}^{T} r(s_t, a_t)$ can be used to update the actions by taking a gradient ascent. For stochastic problems, the reparameterization trick often allows rewriting the next-state distribution $P(s_t, a_t, s_{t+1})$ as a deterministic differentiable function $f(s_t, a_t, \xi_t)$ of exogenous i.i.d. noise $\xi_t$ (Bueno et al. 2019). However, we note that many discrete distributions such as Bernoulli do not support *differentiable* reparameterization.

**PROST** (Keller and Eyerich 2012) is a planning system rooted in the Upper Confidence Bounds applied to Trees (UCT) algorithm. PROST strategically traverses a search tree derived from a factored MDP, efficiently managing the branching factor of chance nodes to reduce the search space size. The algorithm employs a bias parameter that scales with the expected reward of an optimal policy, contributing to convergence. A regret minimization strategy, derived from the UCB1 algorithm, is integrated to enhance performance. PROST also introduces a search depth limitation, prioritizing decisions with more immediate impacts on expected rewards. Additionally, the algorithm excludes actions that lack influence under specific circumstances, leading to a streamlined branching factor. PROST also introduce reward locks for dead-ends and Q-value initialization to avoid random walks, based on a single outcome determinization of the MDP.

**DiSProD** (Chatterjee et al. 2023) is an on-line planner suitable for hybrid domains. Leveraging the stochastic computation graph paradigm, it builds a computation graph capturing an approximation of the distribution over future trajectories and rewards, conditioned on a probabilistic open-loop policy. At each decision step, DiSProD performs a par-

allel search over multiple policies, each optimizing the approximate cumulative reward by differentiating through the computation graph, and then uses the first action from the maximizing policy. DiSProD's novel innovation is its ability to handle mixed discrete-continuous domains by approximating the dynamics with second order Taylor expansions.

## Algorithms and Methodology

**Vectorized Computation Graphs** Our proposed approach to PBBP in RDDL domains, called JaxPlan, leverages the JAX framework to compile the computation graph directly from the lifted RDDL domain description (Frostig, Johnson, and Leary 2018). By working with the lifted problem, it is straightforward to vectorize the computation graph. To illustrate this, consider the conditional probability function

$$cpf(?x, ?y) = \text{sum}_{\{?z:type\}} values(?x, ?y, ?z),$$

where $?x$, $?y$ and $?z$ are free parameters. Working in a depth-first fashion, the $values(?x, ?y, ?z)$ fluent is compiled to a rank-3 tensor of dimensions equal to the number of objects corresponding to types $?x$, $?y$ and $?z$, respectively. Aggregations are compiled to their equivalent tensor operations over one or more axes of the argument (in the example above, a JAX sum operation over the last axis of $values$ tensor). The vectorized just-in-time compilation of RDDL allows JaxPlan to optimize action fluents over a long planning horizon while being efficient in both space and time.

**T-Norm Fuzzy Logic** The novelty of JaxPlan is its ability to handle hybrid continuous-discrete state and action spaces. Specifically, suppose the state factors as $s_t = [s_{t,1}, \dots s_{t,n}]$ where $s_{t,i} = f_i(\text{Pa}(s_{t,i}))$ is a function of the parents $\text{Pa}$ of $s_{t,i}$ in the computation graph, which includes other state and action variables. At the core of its strategy, JaxPlan translates the conditional probability functions $f_i$ into differentiable relaxations, formalized as families of functions $\{\tilde{f}_{i,\tau} : \tau > 0\}$ indexed by some hyper-parameter $\tau$. The variables $\tilde{s}_{t,i} = \tilde{f}_{i,\tau}(\text{Pa}(\tilde{s}_{t,i}))$ and $\tilde{s}_{1,i} = s_{1,i}$ then define a differentiable relaxation of the original model.

Given that Boolean logic in RDDL lacks inherent differentiability, it becomes essential to choose functions $f_{i,\tau}$ that effectively approximate Boolean logic. The novel contribution of JaxPlan is to substitute Boolean operations with t-norms (Hájek 2013). Specifically, a t-norm is a function $T : [0,1]^2 \to [0,1]$ that satisfies four properties: commutativity, monotonicity, associativity, and inclusion of the identity element. For $a, b \in [0,1]$, JaxPlan defines the axioms:

- $a \wedge b \approx T(a, b)$
- $\neg a \approx 1 - a$,

from which other logical RDDL operations can be derived, e.g.:

- $a \vee b \equiv \neg(\neg a \wedge \neg b) \approx 1 - T(1-a, 1-b)$
- $a \implies b \equiv \neg a \vee b \approx 1 - T(a, 1-b)$
- $\forall\{x_1, x_2, \dots x_m\} \equiv x_1 \wedge x_2 \cdots \wedge x_m \approx T(x_1, T(x_2, T(\dots)))$

| Exact RDDL Operation | Differentiable Relaxation |
|---|---|
| $a \wedge b$ | $T(a, b)$ |
| $\neg a$ | $1 - a$ |
| $a \vee b$ | $1 - T(1-a, 1-b)$ |
| $a \implies b$ | $1 - T(a, 1-b)$ |
| $\text{forall}_{\{?p:type\}} x(?p)$ | $\prod_{?p} x(?p)$ |
| $\text{exists}_{\{?p:type\}} x(?p)$ | $1 - \prod_{?p}(1 - x(?p))$ |
| if $c$ then $a$ else $b$ | $c \times a + (1-c) \times b$ |
| $a > b$ | $\text{sigmoid}((a-b)\tau)$ |
| $a == b$ | $\text{sech}^2((b-a)/\tau)$ |
| $\text{signum}(x)$ | $\tanh(x/\tau)$ |
| $\text{argmax}_{\{?p:type\}} x(?p)$ | $\sum_{i=1}^{|type|} i \times \text{softmax}(x/\tau)[i]$ |
| $\text{Bernoulli}(p), \text{Discrete}(p)$ | Gumbel-Softmax |

Table 1: List of differentiable relaxations used in JaxPlan.

- $\exists\{x_1, x_2, \dots x_m\} \equiv \neg\forall\{\neg x_1, \neg x_2, \dots \neg x_m\} \approx 1 - T(1-x_1, T(1-x_2, T(\dots)))$.

For example, the product t-norm defines $T(a, b) = a \times b$, which calculates the logical conjunction exactly when $a$ and $b$ are Boolean.

In addition, JaxPlan approximates the conditional branching statement such as "if $c$ then $a$ else $b$" as

$$f(a, b, c) = c \times a + (1 - c) \times b,$$

which is a differentiable function of its arguments. A popular choice for approximating relational operations, such as $a > b$, $a < b$ and $a == b$ is the logistic sigmoid approximation (Petersen et al. 2021):

$$a > b \approx \text{sigmoid}((a-b)/\tau)$$
$$a == b \approx \text{sech}^2((b-a)/\tau),$$

where $\tau$ refers to the temperature parameter.

Finally, discrete distributions are approximately reparameterized using the Gumbel-Softmax trick (Jang, Gu, and Poole 2016). Formally, samples $z$ from a categorical distribution with density $\{p_i\}_{i=1}^K$ can be approximated by sampling a vector $\xi \in \mathbb{R}^K$ with i.i.d. $\text{Gumbel}(0, 1)$ entries, and computing

$$z = \text{softmax}((\xi + \log p)/\tau)$$

for a temperature hyper-parameter $\tau > 0$.

A more concrete description of the rewriting rules in JaxPlan to facilitate automatic differentiation based on the product fuzzy logic is summarized in Table 1. An interesting property of this set of relaxations is that as $\tau \to \infty$, the overall error in the model relaxation decreases to zero. However, in practice, there is a trade-off between high error in the gradient (for small $\tau$) and vanishing/sparse gradient (for large $\tau$). Our empirical evaluation performs Bayesian hyper-parameter optimization in order to determine the best overall value of $\tau$ for each problem instance, as discussed in a later section.

**Action Parameterization** During optimization, the goal of JaxPlan is to find optimal "soft" actions parameterized as logistic sigmoids, $\tilde{a}_i = \sigma(\theta_i) = 1/(1 + \exp(-w \times \theta_i))$, by

optimizing the real-valued parameters $\theta_i$ using gradient ascent. Here, $w$ is a general hyper-parameter that controls the overall sharpness of the approximation. At test time, since Boolean actions are required, the soft actions $\tilde{a}_i$ are converted to "hard" actions $a_i$ by the rounding operation

$$a_i = \begin{cases} 1, & \text{if } \tilde{a}_i = \sigma(\theta_i) > 0.5 \\ 0, & \text{otherwise.} \end{cases}$$

**Constraint Handling**  Boolean-action domains in IPPC 2011 and 2014 often place non-trivial constraints on the maximum number of non no-op concurrent actions that can be selected in each decision epoch. For example, in the Elevator domain, an agent could open or close the elevator door, or move the elevator up or down in each decision epoch, but cannot perform a combination of the above simultaneously. We assume such constraints can be expressed in the form $\sum_i a_i \leq B$ over all action-fluents $a_i$, for some suitable bound $B$.

In order to ensure constraint satisfaction during optimization, JaxPlan projects the action parameters $\theta = [\theta_1, \theta_2, \ldots \theta_N]$ after each gradient ascent step to the feasible region as follows. First, the soft actions are sorted in descending order to obtain the order statistics $\tilde{a}^{(1)} \geq \tilde{a}^{(2)}, \cdots \geq \tilde{a}^{(N)}$. Next, the $B + 1$-st largest action value determines the amount $\Delta$ by which the soft actions needed to be shifted downwards to satisfy the constraint on the corresponding hard actions, i.e. $\Delta = \tilde{a}^{(B+1)} - 0.5$. If $\Delta > 0$, then all soft action values are shifted downwards to obtain $\tilde{a}_i' = \max(\tilde{a}_i - \Delta, 0)$. Finally, JaxPlan computes the new parameter $\theta_i'$ by inverting the logistic sigmoid $\theta_i' = \sigma^{-1}(\tilde{a}_i')$.

Finally, box constraints for problems in IPPC 2023 are handled simply by clipping any out-of-bounds actions to their valid ranges.

## GurobiPlan

**Planning as Mixed-Integer Nonlinear Programming**
GurobiPlan is an alternative to JaxPlan that could perform better in discrete problems where the model relaxations as defined above could be inaccurate. In summary, this approach involves compiling the return maximization problem into a mixed-integer nonlinear program (MINLP). For example, if $s_{t+1} = As_t + Ba_t$ and $r(s_t, a_t) = c \cdot s_t + d \cdot a_t$ for some appropriately sized matrices $A$, $B$ and vectors $c$, $d$, and a compact (e.g. box-bounded) action space $\mathcal{A}$ the two-stage problem compiles to the mixed-integer linear program:

$$\max_{a_1, a_2 \in \mathcal{A}} \quad c \cdot s_1 + d \cdot a_1 + c \cdot s_2 + d \cdot a_2$$
$$\text{s.t.} \quad s_2 = As_1 + Ba_1.$$

However, GurobiPlan leverages the state-of-the-art Gurobi optimizer to compile RDDL domains directly into MINLPs, which naturally handles a large subset of the nonlinear RDDL operations through piecewise-linear approximations (Gurobi Optimization, LLC 2023).

Additionally, outcomes of relational operators, and some nonlinear functions, are expressed as indicator variables $\delta \in \{0, 1\}$ with suitable constraints imposed on them. For instance, we rewrite $a \geq b$ in terms of a binary variable $\delta$,

constrained as follows:

$$\delta = 1 \implies a - b \geq 0$$
$$\delta = 0 \implies a - b \leq -\varepsilon,$$

where $\varepsilon$ is a small positive constant.

**Constraint Handling**  Another innovation of GurobiPlan is the propagation of tight bounds on variables through interval arithmetic (Hickey, Ju, and Van Emden 2001). To illustrate, for the expression $s' = (s + a)^2$ with $s \in [0, 2]$ and $a \in [0, 1]$, the sub-expression $s + a$ is bounded in $[0, 3]$, and thus $s' \in [0, 9]$. Action preconditions are compiled directly as (possibly nonlinear) constraints on action-fluents using the rewriting rules described above. Together with the box constraints computed through interval arithmetic, and the constraints generated through the rewriting of relational or other operators as in Table 2, this constitutes the constraint set of the complete mixed-integer program.

**Stochastic Variables**  Stochastic variables are not directly compatible with the notion of mixed-integer programming, which is inherently deterministic in nature, thus some form of approximation is required. One option is the sample average approximation, which would require a large sample size in practice and would therefore produce a large nonlinear program that is difficult to solve computationally (Kleywegt, Shapiro, and Homem-de Mello 2002). An alternative and much simpler approach, that is employed in the current implementation of GurobiPlan, is to determinize stochastic samples by replacing them with their mean estimates. For example:

$$b(?p) = \text{Bernoulli}(rate(?p)) \rightarrow b(?p) = rate(?p)$$
$$z(?p) = \text{Normal}(mean(?p), var(?p)) \rightarrow z(?p) = mean(?p).$$

A more complete list of rewriting rules employed by GurobiPlan is summarized in Table 2.

# Empirical Evaluation

## Setting and Metrics

**Benchmark Problems**  The evaluation of JaxPlan and GurobiPlan encompasses the domains of IPPC 2011 (Coles et al. 2012), 2014 (Vallati et al. 2015), and 2023[1]. We note that IPPC 2018 (Geißer 2019) was excluded due to unique, unsupported RDDL language components used exclusively in that iteration of the competition. The evaluation problems benchmark was thus comprised of the 12 domains from IPPC 2011 and IPPC 2014, each with 10 instances, and 7 domains from IPPC 2023, excluding the RecSim domain due to its excessively large combinatorial nature and since no method achieved results significantly better than random behaviour on it. This results in a total of 19 domains and 155 instances. All domains are summarized in the Appendix.

**Baselines**  In the case of IPPC 2011 and IPPC 2014, which were exclusively discrete competitions, we benchmarked the performance of JaxPlan and GurobiPlan against the winner of these two iterations, PROST (Keller and Eyerich 2012).

---

[1]https://ataitler.github.io/IPPC2023/

| Exact RDDL Operation | Gurobi Constraint(s) |
|---|---|
| $\delta = \neg a$ | $\delta + a == 1, \delta \in \{0,1\}$ |
| $\delta = $ if $c$ then $a$ else $b$ | $\begin{cases} c = 1 \implies \delta = a \\ c = 0 \implies \delta = b \end{cases}$ |
| $\delta = a > b$ | $\begin{cases} \delta = 1 \implies a - b \geq \varepsilon \\ \delta = 0 \implies a - b \leq 0 \end{cases}$ |
| $\delta = a \geq b$ | $\begin{cases} \delta = 1 \implies a - b \geq 0 \\ \delta = 0 \implies a - b \leq -\varepsilon \end{cases}$ |
| $\delta = a == b$ | $\begin{cases} \delta = 1 \implies |a - b| \leq 0 \\ \delta = 0 \implies |a - b| \geq \varepsilon \end{cases}$ |
| $\delta = a \neq b$ | $\begin{cases} \delta = 1 \implies |a - b| \geq \varepsilon \\ \delta = 0 \implies |a - b| \leq 0 \end{cases}$ |
| $y = a/b$ | $y \times b == a$ |
| $\delta = \lfloor a \rfloor$ | $\begin{cases} \delta \leq a \\ \delta + 1 \geq a + \varepsilon \end{cases}$ |
| $\delta = \lceil a \rceil$ | $\begin{cases} \delta \geq a \\ \delta - 1 \leq a - \varepsilon \end{cases}$ |
| Bernoulli, Normal... | Determinization |

Table 2: List of rewriting rules using in GurobiPlan.

Meanwhile, the IPPC 2023 iteration introduced continuous and mixed discrete-continuous domain components in the state and possibly action spaces. The winner, DiSProD (Chatterjee et al. 2023), served as the benchmark for planner performance comparison for this set of problems. Complete raw unnormalized results for each evaluated method are provided in the Appendix while this section specifically focuses on the aggregated comparative analysis of the planners on their applicable discrete and mixed discrete-continuous domains. To preempt perceived discrepencies in the the evaluation of JaxPlan vs. DisProd in this work vs. the 2023 IPPC, it is important to note that JaxPlan in this paper is run in replanning mode, whereas in the 2023 IPPC a straight-line plan (no replanning) and reactive policy versions were used.

**Additional Details**    All evaluations were executed on the IPPC 2023 simulation platform, *pyRDDLGym* (Taitler et al. 2022). For PROST, we used the settings as employed in the IPPC 2014 competition, and for DiSProD we used the same settings as employed in the IPPC 2023 competition, except that we modified the algorithms to allow for the increased time budget. For the GurobiPlan and JaxPlan baselines, prior to evaluation, we performed tuning to identify the best hyper-parameter setting for each baseline method on each problem instance. The maximum time budget allowed for tuning was fixed at $2 \times h$ hours, where $h \in \{1, 3, 5\}$ was the maximum allowed time budget per decision epoch (in seconds). For GurobiPlan, we identified the receding horizon $T'$ as the sole key hyper-parameter, whereas for JaxPlan the 5 hyper-parameters and their ranges are defined in Table 5. To identify the best hyper-parameter value for GurobiPlan, we ran a grid search on $\{1, 2, \ldots 20, 22, 24 \ldots 30, 35, 40\}$. To identify the best hyper-parameter value for JaxPlan efficiently within the time budget, we ran Gaussian-process Bayesian optimiza-

tion (Snoek, Larochelle, and Adams 2012), evaluating the average performance of each acquired hyper-parameter setting across 5 independent rollouts of the planner, with random seeds chosen to be different than those used at test time.

**Key Questions**    Our evaluation aims to address three key questions:

1. **Performance Across Domains:** We explore how each method performs in each domain, averaging over the domain's instances. The results are normalized to a scale of [0,1].

2. **Scalability:** We assess how each method fares as instances scale up in size, measured by win rate.

3. **Time Management:** We investigate how each method handles the allocated planning time, examining win rates versus the time per step size.

These questions form the basis of our comprehensive evaluation, providing insights into the strengths and limitations of the considered methods across different dimensions.

## Results and Analysis

We begin with a detailed analysis of results and conclude with a summary of key performance observations at the end of the section.

**Normalized Performance Across Domains**    Following the procedures of the previous IPPC competition, for each instance we compute a normalized score as follows: a score of zero corresponds to the best average return of the no-op and random policy, a score of one corresponds to the best average return across all baselines, and in general the normalized score of a return $R$ is computed as $score = (R - base)/(best - base)$ and clipped to $[0, 1]$, where $best$ is the best return, and $base$ is the maximum of the no-op and random returns. The average score is averaged across all instances for each domain and baseline, and reported in Table 3 for the IPPC 2011 and 2014 domains, and Table 4 for the IPPC 2023 domains.

On the IPPC 2011 and 2014 domains, we see that GurobiPlan achieves the best performance on the AcademicAdvising and CooperativeRecon domains. One possible explanation is that these domains require backwards sequential logical reasoning to identify the optimal path to the goal. For instance, to solve the AcademicAdvising problem, it is necessary to start from the desired course(s) the student wishes to pass (the goal), then pursue their direct prerequisites, the prerequisites of the prerequisites, and so forth. Complex logical constraints on state transitions in these domains renders it more difficult to discover the goal using forward search methods alone – on which JaxPlan and PROST are based. On the other hand, GurobiPlan does relatively poorly on domains where determinization is conjectured to be less effective. For instance, Navigation was designed to provide dead-ends when using most likely outcome determinization, which explains the particularly poor performance of GurobiPlan on this domain.

More generally, with a 5-second time limit per decision epoch, GurobiPlan outperforms JaxPlan on 8 out of the 12

| Domain | GurobiPlan | | | JaxPlan | | | PROST | | |
|---|---|---|---|---|---|---|---|---|---|
| | 1 | 3 | 5 | 1 | 3 | 5 | 1 | 3 | 5 |
| AcademicAdvising | $0.47 \pm 0.31$ | $0.78 \pm 0.26$ | $\mathbf{0.79 \pm 0.26}$ | $0.19 \pm 0.20$ | $0.20 \pm 0.23$ | $0.14 \pm 0.20$ | $0.37 \pm 0.28$ | $0.52 \pm 0.31$ | $0.38 \pm 0.29$ |
| CooperativeRecon | $0.92 \pm 0.06$ | $\mathbf{0.95 \pm 0.07}$ | $0.91 \pm 0.07$ | $0.17 \pm 0.13$ | $0.24 \pm 0.18$ | $0.23 \pm 0.14$ | $0.90 \pm 0.02$ | $0.92 \pm 0.02$ | $0.93 \pm 0.02$ |
| CrossingTraffic | $0.68 \pm 0.13$ | $0.66 \pm 0.13$ | $0.61 \pm 0.16$ | $0.46 \pm 0.13$ | $0.46 \pm 0.14$ | $0.45 \pm 0.14$ | $0.97 \pm 0.04$ | $0.98 \pm 0.02$ | $\mathbf{1.00 \pm 0.00}$ |
| Elevators | $0.88 \pm 0.05$ | $0.87 \pm 0.05$ | $0.88 \pm 0.04$ | $0.29 \pm 0.10$ | $0.36 \pm 0.09$ | $0.36 \pm 0.08$ | $0.97 \pm 0.02$ | $\mathbf{0.99 \pm 0.01}$ | $0.97 \pm 0.01$ |
| GameOfLife | $0.76 \pm 0.24$ | $0.79 \pm 0.20$ | $0.81 \pm 0.14$ | $0.87 \pm 0.07$ | $0.92 \pm 0.04$ | $0.87 \pm 0.08$ | $0.95 \pm 0.02$ | $\mathbf{0.97 \pm 0.01}$ | $0.95 \pm 0.04$ |
| Navigation | $0.20 \pm 0.21$ | $0.34 \pm 0.24$ | $0.37 \pm 0.23$ | $0.56 \pm 0.23$ | $0.61 \pm 0.24$ | $0.74 \pm 0.14$ | $0.81 \pm 0.15$ | $0.87 \pm 0.12$ | $\mathbf{0.96 \pm 0.09}$ |
| SkillTeaching | $0.88 \pm 0.04$ | $0.91 \pm 0.05$ | $0.90 \pm 0.05$ | $0.83 \pm 0.04$ | $0.75 \pm 0.07$ | $0.75 \pm 0.06$ | $\mathbf{0.95 \pm 0.04}$ | $\mathbf{0.95 \pm 0.04}$ | $0.93 \pm 0.03$ |
| SysAdmin | $0.70 \pm 0.09$ | $0.72 \pm 0.09$ | $0.71 \pm 0.10$ | $0.88 \pm 0.06$ | $0.91 \pm 0.08$ | $0.89 \pm 0.07$ | $0.80 \pm 0.09$ | $0.86 \pm 0.08$ | $\mathbf{0.92 \pm 0.04}$ |
| Tamarisk | $0.40 \pm 0.20$ | $0.68 \pm 0.17$ | $0.77 \pm 0.15$ | $0.43 \pm 0.04$ | $0.57 \pm 0.11$ | $0.69 \pm 0.12$ | $\mathbf{0.96 \pm 0.03}$ | $\mathbf{0.96 \pm 0.03}$ | $\mathbf{0.96 \pm 0.04}$ |
| Traffic | $0.65 \pm 0.10$ | $0.84 \pm 0.07$ | $0.84 \pm 0.08$ | $0.68 \pm 0.07$ | $0.70 \pm 0.09$ | $0.64 \pm 0.08$ | $0.98 \pm 0.01$ | $0.99 \pm 0.01$ | $\mathbf{1.00 \pm 0.00}$ |
| TriangleTireworld | $0.21 \pm 0.26$ | $0.32 \pm 0.29$ | $0.36 \pm 0.30$ | $0.12 \pm 0.13$ | $0.11 \pm 0.13$ | $0.16 \pm 0.19$ | $0.81 \pm 0.23$ | $0.90 \pm 0.11$ | $\mathbf{0.99 \pm 0.02}$ |
| Wildfire | $0.29 \pm 0.26$ | $0.55 \pm 0.22$ | $0.69 \pm 0.15$ | $0.94 \pm 0.05$ | $0.95 \pm 0.04$ | $\mathbf{0.97 \pm 0.02}$ | $0.64 \pm 0.20$ | $0.67 \pm 0.20$ | $0.64 \pm 0.22$ |

Table 3: Average normalized performance across instances per domain and per baseline on the combined discrete IPC 2011 and IPC 2014 benchmark problem sets. The best average normalized score achieved per domain is indicated in bold, and intervals represent 95% confidence intervals around the mean score.

| Domain | GurobiPlan | | | JaxPlan | | | DiSProD | | |
|---|---|---|---|---|---|---|---|---|---|
| | 1 | 3 | 5 | 1 | 3 | 5 | 1 | 3 | 5 |
| HVAC | $0.00 \pm 0.00$ | $0.00 \pm 0.00$ | $0.00 \pm 0.00$ | $\mathbf{0.99 \pm 0.01}$ | $\mathbf{0.99 \pm 0.02}$ | $0.98 \pm 0.03$ | $0.96 \pm 0.00$ | $0.96 \pm 0.00$ | $0.96 \pm 0.00$ |
| MarsRover | $0.12 \pm 0.15$ | $0.12 \pm 0.15$ | $0.12 \pm 0.15$ | $\mathbf{0.72 \pm 0.38}$ | $0.62 \pm 0.44$ | $0.41 \pm 0.39$ | $0.24 \pm 0.21$ | $0.24 \pm 0.21$ | $0.24 \pm 0.21$ |
| MountainCar | $0.00 \pm 0.00$ | $0.00 \pm 0.00$ | $0.00 \pm 0.00$ | $0.54 \pm 0.45$ | $\mathbf{1.00 \pm 0.00}$ | $0.88 \pm 0.24$ | $0.48 \pm 0.41$ | $0.48 \pm 0.41$ | $0.77 \pm 0.38$ |
| PowerGen | $0.63 \pm 0.19$ | $0.64 \pm 0.19$ | $0.64 \pm 0.19$ | $\mathbf{0.98 \pm 0.03}$ | $\mathbf{0.98 \pm 0.03}$ | $0.97 \pm 0.02$ | $0.91 \pm 0.05$ | $0.92 \pm 0.05$ | $0.92 \pm 0.05$ |
| RaceCar | $0.00 \pm 0.00$ | $0.00 \pm 0.00$ | $0.00 \pm 0.00$ | $\mathbf{0.20 \pm 0.39}$ | $0.00 \pm 0.00$ | $0.00 \pm 0.00$ | $0.00 \pm 0.00$ | $0.00 \pm 0.00$ | $0.00 \pm 0.00$ |
| Reservoir | $\mathbf{1.00 \pm 0.00}$ | $\mathbf{1.00 \pm 0.00}$ | $\mathbf{1.00 \pm 0.00}$ | $0.99 \pm 0.01$ | $\mathbf{1.00 \pm 0.01}$ | $\mathbf{1.00 \pm 0.00}$ | $0.96 \pm 0.04$ | $0.96 \pm 0.04$ | $0.96 \pm 0.04$ |
| UAV | $0.02 \pm 0.03$ | $0.01 \pm 0.02$ | $0.01 \pm 0.02$ | $0.99 \pm 0.00$ | $0.98 \pm 0.01$ | $\mathbf{1.00 \pm 0.00}$ | $0.97 \pm 0.01$ | $0.97 \pm 0.01$ | $0.97 \pm 0.01$ |

Table 4: Average normalized performance across instances per domain and per baseline on the mixed discrete-continuous IPC 2023 benchmark problem set. The best average normalized score achieved per domain is indicated in bold, and intervals represent 95% confidence intervals around the mean score.

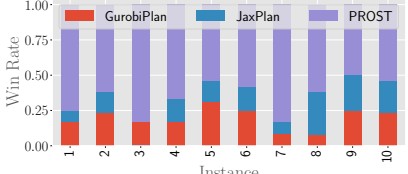

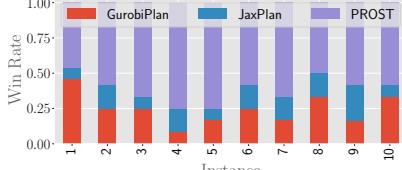

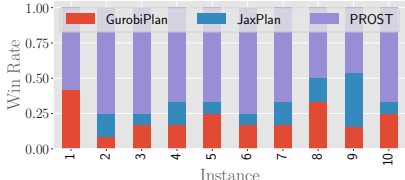

(a) Win rate performance across all 12 domains of IPPC 2011, 2014 vs instance IDs for time step budget of 1 second.

(b) Win rate performance across all 12 domains of IPPC 2011, 2014 vs instance IDs for time step budget of 3 second.

(c) Win rate performance across all 12 domains of IPPC 2011, 2014 vs instance IDs for time step budget of 5 second.

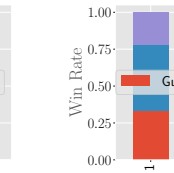

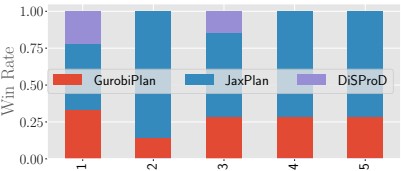

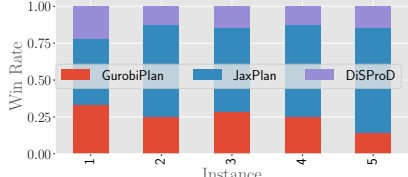

(d) Win rate performance across all 7 domains of IPPC 2012 vs instance IDs for time step budget of 1 second.

(e) Win rate performance across all 7 domains of IPPC 2012 vs instance IDs for time step budget of 3 second.

(f) Win rate performance across all 7 domains of IPPC 2012 vs instance IDs for time step budget of 5 second.

Figure 2: Win rate over all domains vs instance IDs

domains. JaxPlan achieves the best performance on Wildfire, while results on GameOfLife and SysAdmin are comparable to PROST. Meanwhile, JaxPlan achieves the worst performance on AcademicAdvising and TriangleTireworld. We surmise that the performance of JaxPlan on these domains (as well as CrossingTraffic and CooperativeRecon) is hindered by the presence of dead ends, i.e. terminal states that when reached, render the goal state inaccessible during the remainder of the rollout (Guerin et al. 2012; Ng and Petrick 2022). On the other hand, JaxPlan appears to

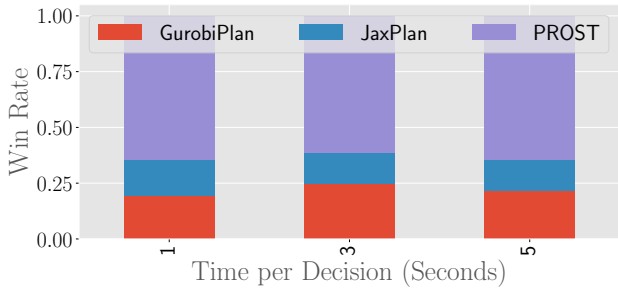

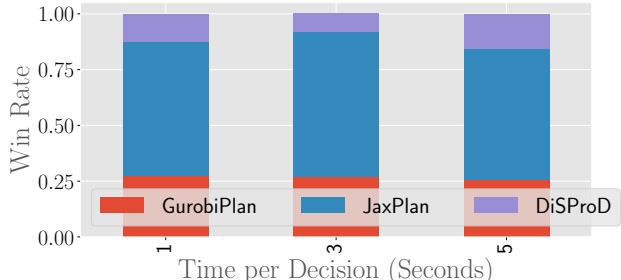

(a) Win rate performance across all 120 instances of IPPC 2011, 2014 vs time budget allocated per decision step.

(b) Win rate performance across all 35 instances of IPPC 2023 vs time budget allocated per decision step.

Figure 3: Win rate performance across all instances vs allocated time per step.

| Symbol | Desc. | Range |
|---|---|---|
| $\sigma$ | Std. deviation of action initializer (Normal) | $[10^{-5}, 10^2]$ |
| $\eta$ | SGD learning rate | $[10^{-5}, 10^2]$ |
| $\tau$ | Sharpness of differentiable relaxation | $[1, 10^5]$ |
| $w$ | Sharpness of soft action parameterization | $[1, 10^5]$ |
| $T'$ | Receding horizon | $\{1, \dots T\}$ |

Table 5: List of hyper-parameters in JaxPlan that were tuned prior to evaluation, together with their corresponding ranges of possible values.

do well when there are multiple high-reward actions available in each state, or at least the presence of a "corrective" action allowing the planner to move from a region of low to high reward, which we typically find in IPPC 2023 domains as we discuss next.

On the IPPC 2023 domains, JaxPlan generally outperforms DiSProD. This new finding does not contradict the results of the IPC 2023 competition, where DiSProD only outperformed the *straight-line* implementation of JaxPlan, not the replanning approach used here. JaxPlan also significantly outperforms GurobiPlan, which fails to make progress on all domains, with the exception of the highly linear Reservoir and PowerGen domains. We suspect the poor performance on the nonlinear domains is at least partially explained by the piecewise linear approximation, which introduces a large number of auxiliary variables. One way to address this issue is to tune the parameters of the piecewise approximation (i.e. number of segments). Another explanation for the poor performance of GurobiPlan is the sparse goal-oriented nature of these problems in particular MountainCar and RaceCar (and to an extent MarsRover and UAV), where full horizon rollouts are required for identifying the goal state. JaxPlan (and its derivative DiSProD) are capable of efficiently planning over a long lookahead horizon within the allocated time budget, since their overall computation cost scales linearly with the rollout horizon.

**Instance Size Scalability** Here, we conduct an evaluation to assess how each method effectively handles problems of increasing size, providing insights into scalability and performance under growing complexities. Across all domains,

the scale of the problem systematically increases with the instance ID. Specifically, instance $i$ exhibits a larger number of objects or a more intricate topology compared to instance $i-1$. This holds true for instances where $i \in \{2, ..., 10\}$ in the case of IPPC 2011 and 2014, and $i \in \{2, .., 5\}$ for IPPC 2023.

To gauge performance, we measure the win rate against instance size for a given time-allocated budget. In essence, the win rate signifies the percentage of successes for each method on a particular instance across all domains. This evaluation metric is applied consistently for instances 1, 2, and so forth, with the measure being relative to the allocated execution time. The evaluation outlined above will be executed for three constant time allocation budgets. This strategic approach allows us to comprehensively measure how each method scales as the time allocated for a planning step size increases. The three time allocation budgets ensure a careful examination of scalability across various planning time constraints. Results are given in Figure 2.

The top row of Figures showcases the performance of JaxPlan, GurobiPlan, and PROST across the domains of IPPC 2011 and 2014. In contrast, the bottom row illustrates the performance of JaxPlan, GurobiPlan, and DiSProD over the 2023 IPPC domains. For both rows, the leftmost Figure illustrates performance over the domains per instance with a time allocation of 1 second per time step. The middle Figure corresponds to a time allocation of 3 seconds per time step, and the rightmost Figure represents a time allocation of 5 seconds per time step.

In the results for the 2011 and 2014 IPPC domains, it is evident that PROST, as a highly tuned method tailored for discrete domains, achieves the highest win rate in the majority of cases. However, specific instances reveal that as the time budget increases, other methods manage to gain the upper hand. For example, in instance 1, GurobiPlan's performance surpasses other methods for step sizes 3 and 5 (see Figures 2b and 2c respectively). An intriguing observation is that while GurobiPlan, as a MINLP method expressing discrete components explicitly, generally achieves better overall results than JaxPlan, there are instances, especially at larger sizes, where JaxPlan outperforms GurobiPlan. No-

tably, in instance 9 and when allocated 5 seconds per step, JaxPlan achieves the best results (see Figure 2c).

Turning to the 2023 IPPC domains, JaxPlan stands out as clearly superior to GurobiPlan and DiSProD. It consistently achieves the best performance across all instance sizes and time budgets, as evidenced in Figures 2d, 2e, and 2f. This demonstrates JaxPlan's robustness and effectiveness in handling the challenges posed by the mixed discrete-continuous nature of the 2023 IPPC domains.

**Time Management and Anytime Performance** Here, we evaluate each method according to its performance in relation to the allocated budget time. The assessment encompasses several time settings in increasing length, to comprehensively understand how each method adapts to varying time constraints.

For every instance, each method underwent execution in three distinct settings, each with a different time budget. The allocation of time was determined on a per-instance basis and was linear with respect to both the horizon and the "step-budget." Specifically, for a problem with a horizon $T$ and a step-budget of $h$, the total allocated time for execution was $T \times h$ seconds per episode. The evaluation was conducted across three step sizes: 1 second per step, 3 seconds per step, and 5 seconds per step.

Performance was quantified through the measurement of win rates, considering all instances across all domains. To maintain clarity and relevance, results from IPPC 2011 and 2014 were distinguished from those of IPPC 2023 due to variations in instance quantity and the methods used for comparison. This separation ensures a focused and controlled analysis of each iteration's results. The results are provided in Figure 3.

We can observe conclusive evidence supporting the assertion that PROST consistently achieves the best results across the domains of IPPC 2011 and 2014, while JaxPlan emerges as the top performer for IPPC 2023. This conclusion holds true across all allocated time step budgets, underscoring the reliability and consistency of these findings.

Notably, an interesting observation from both Figures 3a and 3b is the stability of the win rate distribution among the methods across various time budgets. This consistency implies that the relative performance of each method remains proportionate across different time allocations. Moreover, it is worth noting that all methods demonstrate a capacity to leverage the increased time allocated per step to some degree. Surprisingly, no method establishes a significant lead over the others, indicating a collective adaptability to the additional computational resources provided.

**Results Summary and Key Observations** We now summarize our observations along key evaluation dimensions.

- *Discrete Domains (IPPC2011 & IPPC 2014)*:
  - PROST, being an optimized discrete MCTS method, performs best overall in purely discrete domains.
  - GurobiPlan outperforms JaxPlan on domains with heavy sequential, logical reasoning, but struggles on *Probabilistically Interesting* domains with avoidable probabilistic dead-ends (e.g., Navigation) that are known to provide a failure mode for deterministic replanners (Little and Thiébaux 2007).
  - JaxPlan tends to work better in highly stochastic domains requiring reasoning about many concurrent independent exogenous events where myopic, but stochastic outcome-aware policies can often work well (e.g., Wildfire, SysAdmin and GameOfLife).

- *Mixed Domains (IPPC2023)*:
  - JaxPlan (in replanning mode) performs best.
  - Gurobi's MINLP solver appears to significantly struggle with the horizon required to plan effectively in these challenging mixed discrete-continuous domains.

- *Instance Difficulty and Time Budget*:
  - JaxPlan typically performs better relative to other planners as the instance difficulty increases, particularly in the mixed discrete-continuous instances and very often in the discrete instances.
  - No planner clearly dominates the others as the planning time budget increases.

## Conclusion

In this study, we introduced two distinct optimization-based replanning approaches, JaxPlan and GurobiPlan, designed to address the complexities of planning in mixed discrete-continuous stochastic environments, while also demonstrating the ability to plan in purely discrete scenarios. Our evaluation encompassed a diverse spectrum of problems drawn from past planning competitions, covering both discrete and mixed discrete-continuous domains. In addition to assessing JaxPlan and GurobiPlan, we benchmarked them against the winners of previous competitions, namely PROST and DiSProD, with the overarching goal of establishing baseline approaches and gaining insights into their performance across various scenarios.

Notably, PROST excels in discrete domains, leveraging optimized search-based methods. GurobiPlan, operating as an explicit MINLP method, outperforms JaxPlan in some discrete domains requiring heavy logical, sequential reasoning. However, JaxPlan demonstrated a remarkable ability to surpass GurobiPlan in many discrete instances, especially those that were highly stochastic. A key highlight emerged from our evaluation of the mixed discrete-continuous 2023 IPPC domains, where JaxPlan emerged as the superior performer, consistently excelling across diverse instance sizes and time budgets. This performance underscores JaxPlan's robustness across a range of problem and evaluation types.

Our study positions JaxPlan and GurobiPlan as two unique optimization-based replanning baselines capable of addressing challenges across the spectrum of discrete, continuous, and mixed discrete-continuous problems. These findings not only offer valuable insights into the strengths of each planning method, but also identify key challenges for further advances in the realm of probabilistic planning.

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
