# OpenReview forum: "JaxPlan and GurobiPlan: Optimization Baselines for Replanning in Discrete and Mixed Discrete and Continuous Probabilistic Domains"
_icaps-conference.org/ICAPS/2024/Conference — ICAPS 2024_

### Official Review · Reviewer_besi · 2024-01-19

**Significance And Importance:** 2
**Soundness:** 3
**Novelty:** 2
**Clarity:** 4
**Overall Evaluation:** 2
**Confidence:** 4

**Weaknesses:**

2: No major or minor weaknesses.

**Contributions Of The Paper:**

The paper introduces two novel optimisation-based (re)planning approaches for discrete and mixed discrete-continuous probabilistic domain models: JaxPlan and GurobiPlan. The paper also presents a comprehensive empirical evaluation study over past planning competition benchmarks (IPPCs 2011, 2012, 2014, and 2023) by comparing the proposed approaches JaxPlan and GurobiPlan against two other approaches in the literature (and past planning competitions), i.e., PROST and DiSProD. The study in the paper aims to establish the baseline approaches for (re)planning approaches in discrete and mixed discrete-continuous probabilistic domain models, showing the advantages and disadvantages of each of the evaluated approaches over different domain models with distinct dynamics and characteristics.

**Ethical Considerations:**

(1) Not Applicable: The paper does not have any ethical considerations to address

**Nomination For Best Paper:**

No

**Questions For Authors:**

Q1. As I mentioned in my comment C1, I found myself wondering why you haven't included the approaches of Wu, Say and Sanner (2017) and Bueno et al. (2019) in your empirical evaluation. Is there any reason for that? Please, clarify that.

Q2. You evaluate and assess the approaches in terms of performance, scalability, and time management, but why haven't you considered evaluating the quality of the solution as part of your empirical evaluation?

Q3. Is there any room to improve the proposed approaches JaxPlan and GurobiPlan? You mention that the findings in the paper provide and show the advantages and disadvantages of each of the evaluated approaches, but you don't mention the possible advances that could be made to improve the evaluated approaches. Please, try to address that in your response/rebuttal (and also in the paper).

**Reproducibility:**

5: Code and domains (whichever apply) are already publicly available

**Strengths Of The Paper:**

The paper is well-written and easy to follow. I really like the way the paper is organised and structured. The authors have done a good job at presenting the proposed approaches (JaxPlan and GurobiPlan). I particularly liked the way the authors present the experiments and evaluation section (Empirical Evaluation), it really helps to understand the research findings in their study with respect to the baseline approaches for (re)planning in discrete and mixed discrete-continuous probabilistic domains.

Thus, in my opinion, the key strengths of the paper are as follows:
- A study on baseline planners for (re)planning in discrete and mixed discrete-continuous probabilistic domains (e.g., RDDL);
- The development of two novel optimisation-based (re)planning approaches: JaxPlan and GurobiPlan; and
- A comprehensive empirical evaluation of the two proposed approaches against two other planners (DiSProD and PROST), assessing their performance, scalability, and time management. Such a comprehensive empirical evaluation shows the advantages and disadvantages of each of the evaluated approaches (baselines).

**Weaknesses Of The Paper:**

I see no weaknesses in the paper, but I do have some minor comments and points for consideration, as well as some minor writing issues and typos.

# Minor Comments and Points of Consideration:

C1. I was wondering if the approaches of Wu, Say and Sanner (2017) and Bueno et al. (2019) wouldn't be a nice addition to your experiments (see my question Q1). I think these approaches you could be used in some of the benchmarks (domains and problems) used in the paper.
- https://github.com/wuga214/PAPER_NIPS17_ScalablePlanning_Tensorflow
- https://github.com/thiagopbueno/tf-plan
- https://github.com/thiagopbueno/tf-mdp

C2. Furthermore, personally, I think it would be good (and interesting) to see the results of the straight-line implementation of JaxPlan in the experiments, so we would be able to see how the "novel" JaxPlan (proposed in the paper) has improved compared to the straight-line implementation of JaxPlan.

# Writing Issues and Typos:

W1. "Mixed Discrete and Continuous" -> "Mixed Discrete-Continuous".
W2. "runner up -> "runner-up".
W3. "UCT" -> "Upper Confidence Tree (UCT)".
W4. "Q value" -> "Q-value".
W5. "DiSProD is able to perform a gradient search forward and ..." -> "DiSProD is able to perform a forward gradient-based search and ...".
W6. In Markov Decision Process (MDP), when you say "... solve (1) periodically starting from the current state ...", you mean Equation 1, is that correct?
W7. "i.e." -> "i.e.,".

---

> ### Author Rebuttal · Authors · 2024-01-28
>
> We thank the reviewer for their supportive comments and questions.
>
> **C1, Q1:** In general, JaxPlan can be seen as an extension of the Tensorflow-based planner in Wu, Say and Sanner (2017), which we will call "TensorPlan" here.  While the TensorPlan paper claims to handle discrete state, such problems are strictly non-differentiable and TensorPlan did not provide any practical method for handling (e.g., relaxing) discrete states -- used in all IPPC domains evaluated in this paper -- as also evidenced by the fact that the TensorPlan paper only experimented with continuous state and action domains.  One of JaxPlan's main contributions is to handle discrete state/action spaces, non-differentiable dynamics, and nontrivial action constraints necessary for the IPPC domains.
>
> Interestingly, JaxPlan is technically compatible with the Deep Reactive Policy (DRP) policy architecture of Bueno et al (2019), however the main reason for not including it was to compare different baseline algorithmic methodologies (MCTS used in PROST, Gurobi MINLP optimization, Jax Gradient-based optimization) head-to-head under the *same replanning conditions* in one paper.
>
> **C2:** The straight-line implementation of JaxPlan (i.e., no replanning) that served as a baseline for the 2023 IPPC and was [outperformed by DisPRod](https://ataitler.github.io/IPPC2023/results/IPPC2023_results.pdf).  Since JaxPlan (with replanning) typically outperformed DisPRod in the paper, this would not have added significantly to the analysis.  And we believe this makes sense because JaxPlan with replanning should match or outperform JaxPlan without replanning (that does not get to recover from its mistakes).  We will add this interesting discussion on revision.
>
> **Q2:** We focused our analysis of win rates and normalized returns averaged across instances, following the past IPPC competitions. We also reported the per-instance returns in the Appendix. However, your question concerning plan quality is interesting. The question is how to properly evaluate plan quality, since a majority of the instances are challenging for the state-of-the-art solvers and we do not have optimal plans to compare to.  If the reviewer has any suggestions, we would welcome them.
>
> **Q3:** These are excellent questions and we would ask you to see the response to Q1 from reviewer i7BH who asked the same question.  We will add this discussion on revision.
>
> **Text revision comments:** thank you and we will revise accordingly.

---

### Official Review · Reviewer_i7BH · 2024-01-22

**Significance And Importance:** 3
**Soundness:** 3
**Novelty:** 3
**Clarity:** 3
**Overall Evaluation:** 2
**Confidence:** 3

**Weaknesses:**

2: No major or minor weaknesses.

**Contributions Of The Paper:**

Thanks to the authors for their responses to all reviews.  I remain convinced that the paper should be accepted, and look forward to seeing the promised revisions in the final version.

Original review:

This paper presents an extensive discussion and evaluation of the implementations, performance characteristics, and relative strengths and weaknesses of two different systems for mixed discrete/continuous probabilistic planning:  JaxPlan and GurobiPlan.  The two planners are evaluated on both discrete probabilistic planning problems (IPPC 2011 and 2014) and hybrid probabilistic planning problems (IPPC 2023), including benchmarking against the planners that won those competitions: Prost for 2011, 2014, and DiSProD for 2023.

**Ethical Considerations:**

(1) Not Applicable: The paper does not have any ethical considerations to address

**Nomination For Best Paper:**

No

**Questions For Authors:**

1.  What challenges for further advances are identified (or implied) by the results presented here?

**Reproducibility:**

5: Code and domains (whichever apply) are already publicly available

**Strengths Of The Paper:**

While the paper does not introduce much in the way of new results in the form of, say, planning techniques, it does provide an extensive analysis of the relative strengths and weaknesses of several different approaches to probabilistic planning in both discrete and hybrid domains.  That analysis is backed up by extensive experimental results, with some attempt to describe those results in terms of the different technical approaches taken in the different planners.

**Weaknesses Of The Paper:**

The only real problem I had with the paper was the presentation of graphical data.  Overlaying the legend on top of the graphs themselves obscures the plots in confusing ways.  See for example Figure 3.a, where it appears as though the Win Rate does not sum to 1.00.  That is an artifact of a bad layout, not a problem with the data or the plot itself.
As a suggestion: in the second full paragraph in the second column on p. 7, consider replacing the term "Figures" with "graphs."

---

> ### Author Rebuttal · Authors · 2024-01-28
>
> We thank the reviewer for their supportive comments and question.
>
> Good point, we will move the legends outside of the graphs.
>
> **Text revision comment:** thank you and we will revise accordingly.
>
> **Q1:** There are many exciting future directions, but we will focus on the most notable ones from our analysis:
> 1. GurobiPlan and JaxPlan both struggle on probabilistically interesting problems with dead ends (according to IPPC 2011 and 2014 results), which should incorporate extensions from other non-deterministic planners intended to resolve these issues in future work.
> 2. GurobiPlan does not scale well to long planning horizons, unlike JaxPlan, which is necessary to succeed on some sparse goal-oriented problems in IPPC 2023 (MountainCar and RaceCar are notable examples).  Hence we would like to explore methods for extending GurobiPlan's planning horizon.
> 3. JaxPlan's performance is sensitive to the quality of the approximation of discrete state and action components, so future work should explore the rich design space of t-norm approximations and better (online) tuning of key hyper-parameters.

---

### Official Review · Reviewer_N9DJ · 2024-01-22

**Significance And Importance:** 2
**Soundness:** 4
**Novelty:** 3
**Clarity:** 4
**Overall Evaluation:** 2
**Confidence:** 4

**Weaknesses:**

1: Minor weaknesses that are easily fixable.

**Contributions Of The Paper:**

=== POST-REBUTTAL ===
I thank the authors for their insightful rebuttal comments and I still strongly support to accept this paper. I hope the authors will revise the paper as they promise in the rebuttal, especially concerning the explanation why JaxPlan does not excel in probabilistically interesting domains contrary to the intuition.


=== ORIGINAL REVEW ===

- The paper describes 2 algorithms that can solve RDDL domains with discrete states or mixed discrete/continuous states: JaxPlan and GurobiPlan, the former being a replanning version of the baseline planner of IPPC 2023. JaxPlan is based on a JAX compilation of the lifted RDDL domain which then allows for policy gradient optimisation making use of the JAX framework, while GurobiPlan relies on a MINLP formulation of the problems then solved by Gurobi.

- The authors compare their 2 algorithms against the winners of IPPC 2011 and 2014 for discrete domains (PROST) and of IPPC 2023 for mixed discrete/continuous domains (DiSProD).

- In overall, the experiments show that PROST outperforms JaxPlan and GurobiPlan on most discrete domains, but a few exceptions in favour of JaxPlan or GurobiPlan, while JaxPlan systematically outperforms DiSProD in mixed discrete/continuous domains (the version of JaxPlan presented in this paper being used in replanning mode).

**Ethical Considerations:**

(1) Not Applicable: The paper does not have any ethical considerations to address

**Nomination For Best Paper:**

No

**Questions For Authors:**

- "FF-Replan is excluded from our evaluation here due to its specificity to PPDDL and challenges in adapting to RDDL domains featuring concurrency or continuous components" => I am puzzled: since PPDDL is less expressive than RDDL, yielding less combinatorial action effects in general, I would expect RDDL planners to be more efficient than pure PPDDL planners. Also, there is no technical reason to prevent RDDL planners from being compared against PPDDL planners on the less expressive set of problems that can be expressed in PPDDL only (and eventually translated to RDDL). Can you please convince me of the opposite? The same argument holds for PROST which is mentioned afterwards in the same paragraph. (TL;DR : "if you can move mountains you can move molehills")

- "b(?p) = Bernoulli(rate(?p)) -> b(?p) = rate(?p)" => does it mean that the b boolean variable becomes a float variable after the mean estimate transformation?

- "The maximum time budget allowed for tuning was fixed at 2 × h hours, where h in {1, 3, 5} was the maximum allowed time budget per decision epoch (in seconds)" => I'm lost: are we speaking about hours or about seconds?

- When Gurobi is used in the literature to solve scheduling problems, each variable is usually indexed by time. Is it the same of the MINLP model used by GurobiPlan?

- JaxPlan appears to perform not so well in probabilistically interesting domains (e.g. AcademicAdvising and TriangleTireworld) whereas its JAX encoding handles probability distributions. How do you explain this? Is it due to a too loose approximation of the probability distributions as the Gumbel-Sotfmax transformation?

- "JaxPlan typically performs better relative to other planners as the instance difficulty increases" => The claim being made also for IPPC 2011 and 2014 domains, I don't see how Figure 2/(a)-(c) supports this statement. Can you please elaborate?


=== General comments ===

- "PROST also introduce [...]" => introduceS

- "DiSProD’s novel innovation" => "novel" and "innovation" are redundant

- "JaxPlan achieves the best performance on Wildfire, while results on GameOfLife and SysAdmin are comparable to PROST." => well, honestly, not really for GameOfLife...

**Reproducibility:**

5: Code and domains (whichever apply) are already publicly available

**Strengths Of The Paper:**

- I found the JaxPlan and GurobiPlan algorithms very interesting and well described, with sufficient details to understand how they globally work in practice.

- The paper is globally well written and pleasant to read.

- The comparison of the different approaches is interesting and useful in the sense that I feel that I gained knowledge on which algorithms I should choose to solve a given RDDL domain based on its distinctive feature (like for instance being fully discrete or mixed discrete/continuous; or exhibiting complex logical constraints or not).

- The JaxPlan planner in replanning mode appears to be the most efficient planner to solve mixed discrete/continuous RDDL domains.

**Weaknesses Of The Paper:**

I find that Figure 3 and the subsection "Time Management and Anytime Performance do not bring fundamentally new information with regard to Figure 2. It is certainly a different view on the same results, yet redundant in my opinion. Also, I felt that Figure 3 was maybe exaggeratedly made bigger to fill the space, but I might be wrong. I would have preferred instead more details about the JaxPlan and GurobiPlan planners, like for instance their pseudo-codes and formal definitions of their model transformations.

---

> ### Author Rebuttal · Authors · 2024-01-28
>
> We thank the reviewer for their supportive comments and questions.
>
> Indeed, Figure 3 is just a complementary view of the main results in Figure 2 to show performance explicitly vs. time per decision while aggregating over instance difficulty.  We will reduce the size to fit in more method and discussion details.
>
> **Q1:** It seems possible to directly translate at least Navigation and AcademicAdvising to PPDDL to compare to FF-Replan.  If the reviewer believes other domains can be directly translated, please let us know.  Just for reference, the paper on [Glutton](https://aiweb.cs.washington.edu/ai/planning/papers/GLUTTON.pdf) by Kolobov et al (ICAPS-12) that placed second to PROST in IPPC 2011 provides an excellent discussion of why the winners of pre-2011 PPDDL competitions such as FF-Replan are not generally appropriate for RDDL domains in IPPC 2011.  We will cite this paper and its discussion on revision.
>
> **Q2:** Correct, the boolean RV is relaxed into a deterministic float.
>
> **Q3:** If the decision budget is 3 seconds per time step, then the total time allocated to tuning is 6 hours. This ensures that we can compare baselines using their (near)-optimal hyper-parameter settings.
>
> **Q4:** Indeed, all variables in the Gurobi compilation have a time index up to the planning horizon.  We will clarify on revision.
>
> **Q5:** You ask an excellent question about JaxPlan's performance on probabilistically interesting domains and we will add more discussion on revision.  JaxPlan uses i.i.d. sampling of reparameterized stochastic variables at each time step, hence if there are a sequence of low probability transitions to a dead-end, JaxPlan may sample all of the transitions as avoiding the dead-end, even though the cumulative probability of diverting to the dead-end is high.  Thus, JaxPlan suffers from the same drawbacks that plague many determinized replanners.  We will mention that future work on JaxPlan should look at prior extensions of replanners intended to resolve such issues.
>
> **Q6:** This qualitative observation is based on the fact that for simple instances e.g., 1-3 JaxPlan has minimal amount of wins, even zero for instance 3 Figure 2a and instance 1 Figure 2c, while it has more wins and even on instances 7-10 and in instance 9 it has the most across the planners in Figure 2c and competitive results with PROST in the others.
>
> **General comments:** thank you and we will revise accordingly.

---

### Meta-Review · Area_Chair_QNMZ · 2024-02-05

**Recommendation:** Accept (Oral)
**Confidence:** 5

**Metareview:**

The paper introduces two replanning approaches to probabilistic planning, presented as potential baselines in an evaluation setting.

The two approaches rely on distinct optimization settings and are clearly presented and interesting in themselves. The authors perform a comprehensive empirical study on IPC benchmarks and against winners of past competitions, identifying several insights regarding the performance of the proposed solvers.
Importantly, the authors identify key weaknesses in more established probabilistic planning algorithms that may be leveraged in driving future research in the area.

**Ethical Considerations:**

(1) Not Applicable: The paper does not have any ethical considerations to address